# Polyacrylonitrile-Derived Carbon Nanocoating for Long-Life High-Power Phosphate Electrodes

Eugene E. Nazarov [1],*[iD], Oleg A. Tyablikov [1], Victoria A. Nikitina [1], Evgeny V. Antipov [1,2] and Stanislav S. Fedotov [1],*

[1]   Skoltech Center for Energy Science and Technology, Skolkovo Institute of Science and Technology, Bolshoi Boulevard 30 bld. 1, 121205 Moscow, Russia
[2]   Department of Chemistry, Lomonosov Moscow State University, 119991 Moscow, Russia
*   Correspondence: eugene.nazarov@skoltech.ru (E.E.N.); s.fedotov@skoltech.ru (S.S.F.)

**Abstract:** Carbon-coating proved an efficient and reliable strategy to increase the power capabilities and lifetime of phosphate-based positive electrode materials for Li-ion batteries. In this work, we provide a systematic study on the influence of polyacrylonitrile-(PAN)-derived carbon coating on electrochemical properties of the nanosized Li-rich $Li_{1+\delta}(Fe_{0.5}Mn_{0.5})_{1-\delta}PO_4$ (Li-rich LFMP) cathode material, as well as the characterization of carbon-coated composites by means of Raman spectroscopy for the determination of carbon graphitization degree, DF-STEM and STEM-EELS for the estimation of carbon layer thickness, uniformity and compositional homogeneity of the conductive layer respectively, and impedance spectroscopy for the determination of charge transfer resistances of the resulted composite electrodes in Li-based cells. Using PAN as a carbon coating precursor enables significantly enhancing the cycling stability of Li-rich LFMP/C compared to those conventionally obtained with the glucose precursor: up to 40% at high current loads of 5–10C retaining about $78 \pm 2\%$ of capacity after 1000 cycles. Varying the PAN-derived carbon content in the composites allows controlling the electrochemical response of the material triggering either a high-capacity or a high-power performance.

**Keywords:** Li-ion batteries; triphylite-type cathodes; surface modification; carbon coating; PAN

## 1. Introduction

In the quest for implementing higher energy density, longer-life and safer Li-ion batteries targeted to electromotive transportation and large-scale grid storage systems, significant attention of the research community was paid to the triphylite-type positive electrode materials, $LiMPO_4$ (M = Fe, Mn, Co, Ni). Specifically, Mn-containing $LiFe_xMn_{1-x}PO_4$ (LFMP) solid solutions received the largest interest from the industry among all the counterparts due to the minimal increase in the cost and improved specific energy over the highly developed and commercialized $LiFePO_4$, thus facilitating the deployment of LFMP to the market.

In order to enhance the specific power characteristics of the $LiFe_xMn_{1-x}PO_4$ materials, several approaches were proposed, including nanostructuring [1], modification of the defect structure [2], and carbon-coating [3], which are usually applied in ensembles. In particular, nanosizing aims to reduce the diffusion paths and could affect the de/insertion mechanism; this involves tuning the defect structure results in amplifying $Li^+$ diffusion coefficients, increasing the diffusion network dimensionality [4,5], and extending the solid solution region of the de/intercalation mechanism [2,6,7]. Casting a conductive coating on the particle's surface enhances the grain-boundary electronic conductivity, ensuring much lower polarization.

Besides elevating electronic conductivity, the use of carbon coatings opens up further opportunities for upgrading cathode materials. It also provides a higher mechanical and

chemical stability of the electrode composite. Furthermore, for the Mn-containing electrode materials with their partial dissolution of $Mn^{2+}$ in the electrolyte [8,9], it was possible to reduce the leaching of both Fe and Mn by increasing the thickness of carbon coating without deterioration of the electrochemical performance [9]. One more positive aspect of forming the conductive carbon layer is the prevention of materials particles from coalescence or infusion at high-temperature treatment. This is especially crucial for nanosized LFMP-type cathodes designed and prepared to provide fast $Li^+$ diffusion. Additionally, the carbon coating impedes the electrolyte decomposition catalyzed by available active d-metal centers at the surface, thus prolonging the battery operation and lifetime [10].

Typically, materials particles are coated with a conductive carbon layer derived from the organic precursor decomposed during high-temperature annealing. Such organic precursors could be divided into two major groups: monomer (glucose [11], sucrose [10], citric [12], or ascorbic [13] acids) and polymer compounds. As polymer compounds, a variety of compositions is utilized: polyacrylonitrile (PAN) [14,15], polyaniline (PANI) [16], resorcinol–formaldehyde [17], etc. The common trend for the use of polymers is that the properties of the carbon layer, such as uniformity and thickness, could be adjusted in the polymerization process or by mixing conditions of the precursor and an active material [16]. Attempts to describe the influence of different carbon sources on electrochemical properties of triphylites were made in [18], where various compounds were considered as carbon-coating precursors for LFMP: carbon black, citric acid, polyvinylpyrrolidone (PVP), etc. As comparison parameters, values of electronic conductivity and/or charge transfer resistances and output specific capacities, as well as cycling stability, are usually noted. Among the chosen organics, the maximum discharge capacity and the lowest charge transfer resistance values were observed for the PVP-derived carbon-coated LFMP/C due to the specific adsorption of PVP on the hydroxyl-terminating surface layer of LFMP [18]. Thus, a precursor for carbon coating application should have a good affinity to an electrode material surface in order to create a uniform and homogeneous conductive layer.

Among the polymer-type precursors, PAN is worth mentioning, as it does not only attribute to the electronic conductivity enhancement of the resulting carbon-containing composite, but also provides own decent $Li^+$ conductivity of more than $10^{-4}$ S·cm$^{-1}$ when used in ceramic/polymer electrolytes for solid-state batteries. Due to the presence of cyano-groups, PAN displays a good affinity to metals ions, which could be further improved by modifying the polymer [19], leading to a more uniform and conductive carbon layer. Another supportive evidence for a high PAN affinity to transition metal oxides surface could be found in [20], where PAN was applied as a binder for electrodes preparation. Compared with a conventional PVDF binder, the capacity retention of lithium manganese cobalt nickel oxide with PAN as a binder enlarged by 7% after 300 cycles at 5C rate [20]. Despite the adsorption modifications, PAN could be produced in a cyclic form by a pre-oxidation in air, which noticeably boosts the conductivity of the final composite, resulting in a higher cycling stability with more specific capacity being delivered at elevated current densities [21]. The utilization of PAN for polyanion cathodes was described in [22], where PAN-derived carbon coating enables to achieve specific capacity values close to the theoretical one for the $NaVPO_4F$ cathode. Applying PAN as a carbon coating precursor for $KTiPO_4F$ cathode material made it possible to preserve $Ti^{3+}$ oxidation state due to the absence of chemically bonded oxygen in the polymer [15]. Thus, PAN coatings could be suggested for other cathode materials containing easily oxidized transition metals such as $V^{3+}$, $Fe^{2+}$, etc. As a carbon coating precursor, PAN is applied not only in technology of cathode materials, but also for anode materials. PAN-derived carbon coating was successfully implemented to yield high output capacity and cycling stability of a $SnO_2/C$ composite [23]. The resulting carbon matrix facilitates the electronic transport and also provides mechanical stability of the composite during cycling [23]. The close results were obtained in the technology of the GeP anode preparation, where a mixed carbon source composed of PAN and carbon nanotubes was applied [24]. The application of the complex carbon coating made it possible to reduce both charge transfer resistance of the composite and capacity fade during long

cycling [24]. Moreover, PAN could be extracted from spent fabric rendering the polymer an abundant and cheap precursor for conductive carbon coating preparation [25].

Another approach allowing tailoring the surface properties of the obtained carbon coating is using MOFs (metal organic frameworks) as a carbon coating precursor [26]. As is the case for PAN technology, it is possible to control the electronic conductivity of the carbon layer by introducing additives or engineering the chemical structure of the organic framework [27,28]. Currently, the described approach is applied mostly for laboratory applications due to a complicated synthesis procedure of MOF materials. Usually, the process requires a number of steps, including solvothermal treatment, careful solvent extraction from the porous structure, and a cumbersome drying process [29].

Taking into account the described research made on PAN for the implementation in the area of electrode materials, one could consider the polymer as a promising carbon coating precursor for long-life electrode materials, as it has a good affinity to different 3d-metal-containing surfaces, thus contributing to the formation of a thin but uniform carbon coating that might result in a better cycling stability, reduced dissolution of transition metals ions in the electrolyte, and higher C-rate capabilities. The absence of oxygen in chemical composition of PAN makes it suitable for application as a coating for sensitive material containing transition metals with intermediate oxidation state. Utilization of recycled PAN from spent fabric could be an additional benefit of applying the polymer in the technology of electrode materials.

Herein, we report on the preparation, study, and application of a PAN-derived only carbon nanocoating for the extension of the high-power capabilities and cycling stability of the recently designed Li-rich LFMP with extra Li at the 3d-metal sites. The obtained carbon-coated composite materials, Li-rich LFMP/C, exhibit an increased capacity retention of up to $78 \pm 2\%$ after 1000 cycles under a combined 5C/10C cycling protocol in Li half-cells compared with only $43 \pm 3\%$ for a glucose-derived carbon-coated Li-rich LFMP. The comparative study on charge transfer resistances demonstrates a much faster kinetics of the composites with PAN-derived coatings during both Fe and Mn redox transitions over the ones with the glucose-based coating.

## 2. Materials and Methods

Li-rich LFMP cathode materials were produced via a solvothermal treatment of a lithium phosphate precursor. The synthesis scheme was close to the one published earlier in our work [2]. Prior to the synthesis, the water content for all crystal hydrates was determined by a thermogravimetric (TG) analysis. For a typical synthetic route, $LiOH \cdot H_2O$ (Komponent reactive, 99%, 45.4 g) was dissolved in 144 mL of water–ethylene glycol mixture (30:70 by vol.), followed by the addition of 25.4 mL $H_3PO_4$ (Komponent reactive, 85% aqueous solution) with the formation of a white $Li_3PO_4$ precipitate. In a separate beaker, ascorbic acid (Ruskhim, 99%, 2.9 g), $FeSO_4 \cdot 7H_2O$ (Komponent reactive, 98%, 50.3 g) and $MnSO_4 \cdot H_2O$ (Komponent reactive, 98%, 30.4 g) were completely dissolved in 72 mL of deionized water. The resulting solution was poured into the lithium phosphate suspension, transferred to a 600 mL stainless steel autoclave (Parr instruments), purified with Ar, and sealed. The autoclave was heated up to 190 °C at a 1.5 K/min heating rate and kept for 1 h. A brown suspension was collected by centrifugation, washed several times with deionized water and ethanol, and dried in a vacuum oven at 70 °C overnight. Li-rich LFMP/C using PAN (Toyo Japan) as a carbon coating precursor was produced by the following scheme (Figure 1):

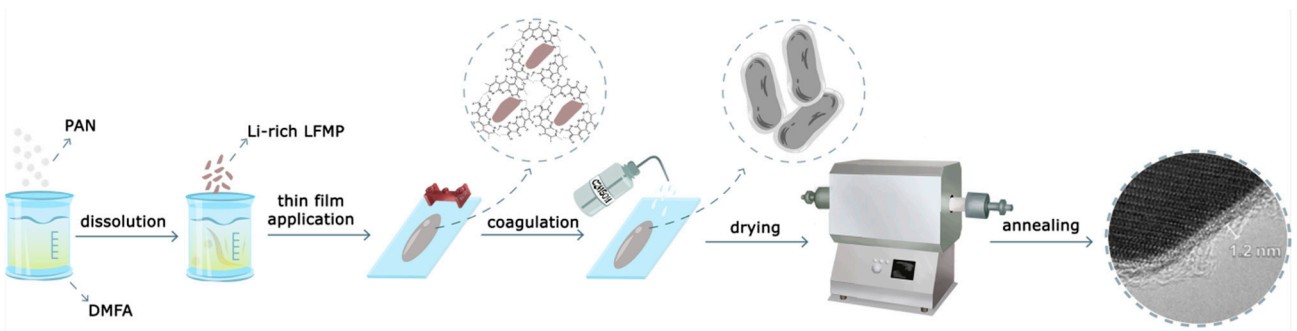

**Figure 1.** Schematic representation of PAN-nanocoating application technology.

First, a suitable amount of PAN (2, 4, and 8 wt. % calculated to the total mass of the final composite) was dissolved in dimethylformamide (DMFA), and the samples are denoted as PAN-2, PAN-4, and PAN-8 correspondingly, followed by the addition of the dried active material, and the description of mass ratios is represented in Table S1. The average molar mass of PAN is ~5000 g/mol, and the powder consists of spherical particles with a diameter of 150–200 nm (Figure S1).

The resulting slurry was magnetically stirred for 1 h and applied on a glass surface, then rinsed with ethanol in order to activate the PAN coagulation process. The obtained thin film was washed with ethanol in order to eliminate the DMFA excess and dried under dynamic vacuum at 60 °C followed by annealing at 650 °C for 12 h, and the heating rate was 3 K/min. As a reference sample, a Li-rich LFMP/C composite with glucose as a carbon source was obtained. For that, the active material was mixed with 10% mass of glucose dissolved in ethanol and stirred for 1 h. The slurry was dried at 80 °C under dynamic vacuum and annealed at 650 °C for 12 h, with the heating rate being 3 K/min.

Phase purity and cell parameters were extracted from a PXRD experiment using Bruker D8 Advance diffractometer (Cu-K$\alpha$ radiation, $\lambda_1$ = 1.54056 Å, $\lambda_2$ = 1.54439 Å) equipped with an energy-dispersive LYNXEYE XE position-sensitive detector. Rietveld refinement was performed using the GSAS II software package [30].

The morphology characterization was conducted using a Quattro S scanning electron microscope (ThermoFisher, Waltham, MA, USA).

The carbon content in the resultant composites was measured by a thermogravimetric analyzer Netzsch STA 449 F3 Jupiter under Ar:O$_2$ (80:20) flow at a 5 K/min heating rate in the corundum crucible with lids. At first, the mass increase was measured on the pure material without carbon coating precursor, and the obtained value was further used for the precise determination of the carbon content.

Fourier transform infrared (FTIR) spectra of the annealed material and graphitized carbon coating precursors were collected with an ALPHA II compact FT-IR spectrometer. The spectra were recorded in the 4000–400 cm$^{-1}$ range with 2 cm$^{-1}$ resolution and averaging 3 scans. The reproducibility was checked by probing different spots of the same powder sample.

The graphitization degree (the intensity ratio of D over G band) of the composites was probed by means of a Raman spectroscopy experiment using a DXR3xi Raman Imaging Microscope (Thermo Fisher). The laser intensity was set at 0.7 mW, and the exposition time was 1 Hz with 40 frames resolution.

The uniformity of a carbon coating was analyzed by transmission electron microscopy (TEM). The samples for TEM investigation were prepared in air by crushing the crystals in a mortar in acetone and depositing drops of suspension onto holey copper grids with a Lacey/carbon support layer. TEM images, energy-dispersive X-ray (EDX) spectra, and maps were taken on an aberration-corrected Titan Themis Z transmission electron microscope, equipped with a Super-X EDX detection system and operated at 200 kV. Electron energy loss spectroscopy (EELS) data were recorded in a STEM mode with a Gatan Quan-

tum ER965 spectrometer; the energy resolution of the zero-loss peak was 1.1 eV. For every composite, a set of 20 particles was chosen for the experiment.

The suspension for electrodes preparation was produced by mixing the active material (90%), PVDF binder (5%), and carbon black (5%) (Xiamen Tob New Energy Technology Co., Ltd., Xiamen, China) with a N-methyl pyrrolidone (NMP) solvent (Sigma Aldrich, Saint Louis, MO, USA) in a SPEX ball-milling machine 800. The resulting suspension was applied onto a carbon–aluminum foil using a doctor blade automatic film applicator with the thickness of 150 μm. The electrode foil was calendered, and the electrodes with a total area of 2.25 cm$^2$ were cut and dried in a vacuum oven at 120 °C overnight. For galvanostatic cycling, CR2032 coin cells with Li metal as an anode were used. A typical commercial electrolyte consisting of 1M LiPF$_6$ in EC:DEC (1:1 by vol.) was used. Galvanostatic cycling was carried out in the potential range of 2.5–4.3 V vs. Li/Li$^+$ using a BTS tester (Neware). Charge transfer resistance values were determined from impedance spectroscopy measurements using three electrode cells with lithium as counter and reference electrodes. The frequency range was 10 mHz–100 kHz. Prior to the impedance spectroscopy experiments, cyclic voltammograms (CV) were recorded at 50 μV/s in the potential range 3.0–4.3 V vs. Li/Li$^+$ to estimate the phase transition potentials. The impedance spectroscopy measurements were performed in 5 mV steps in the potential ranges 3.415–3.44 V vs. Li/Li$^+$ for the Fe$^{2+/3+}$ transition and 3.964–4.018 for the Mn$^{2+/3+}$ transition, which correspond to the solid solution regions of Li$^+$ de/intercalation during the charge process. The width of the solid solution regions was determined from the analysis of the shape of current transients, which were recorded after each potential step for 30 min or until the current approached the background values. The current transients recorded in the solid solution and two-phase ranges are presented in Figure S6. Impedance spectra were analyzed using MEISP software [31]. The equivalent circuit, which was used to fit the spectra, is shown in Figure S7.

## 3. Results

All Li-rich LFMP/C samples in this work are single-phase assigned to the orthorhombic *Pnma* space group (Figure 2) with unit cell parameters as well as experiment details presented in Table 1. A precise description of the Li-rich LFMP material's crystal structure can be found in our previous work [2]. The morphology of the synthesized carbon-containing composite materials is characterized by platelet-like primary particles with a particle size in the range of 50–100 nm Figure 2. The thickness of the particles does not exceed 50 nm.

**Table 1.** Unit cell parameters determined from the PXRD refinement.

| Space Group | *Pnma* |
|---|---|
| *a*, Å | 10.3847(4) |
| *b*, Å | 6.0502(6) |
| *c*, Å | 4.7196(7) |
| *V*, Å$^3$ | 296.538(5) |
| Z | 4 |
| GOF | 0.11 |
| $R_{exp}$, % | 0.94 |

TG experiments were conducted to estimate the total carbon content in the samples. For the carbon-coated samples, the total mass change during the TG experiments arises from two processes: oxidation of d-metals and carbon burning. The experimental TG curves are presented in Figures S2 and S3a–d. The difference between expected and experimentally determined carbon contents is given in Table 2, which does not exceed 0.4%.

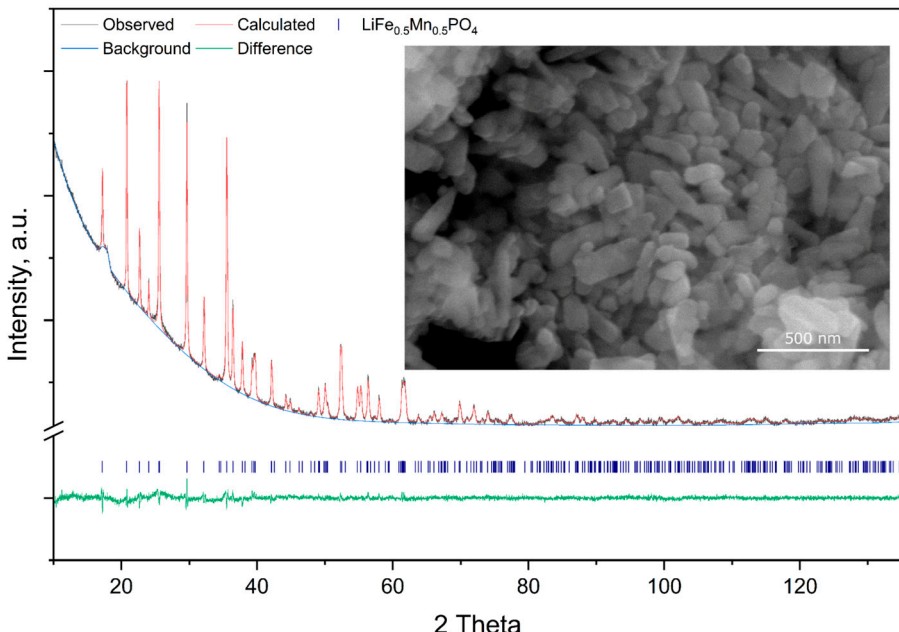

**Figure 2.** Rietveld refinement of the XRD pattern of the as-prepared LFMP. Inset: SEM image of the as-prepared material.

**Table 2.** Mass change and determined carbon content values for Glu and PAN samples.

| Sample | Mass Change, % | Carbon Content, % Experimental | Carbon Content, % Expected | Graphitization Degree |
|---|---|---|---|---|
| Li-rich LFMP pristine | +2.36 | - | - | - |
| PAN-2 | −0.3 | 2.4 | 2 | 1.4 |
| PAN-4 | −1.94 | 4.2 | 4 | 1.3 |
| PAN-8 | −5.8 | 7.9 | 8 | 1.2 |
| Glu | −1.37 | 3.8 | 4 | 0.9 |

The first visible DTA signal and slight mass loss is connected with the physically adsorbed water evaporation. The next sharp peak at 300 °C could be assigned to the $Fe^{2+}$-to-$Fe^{3+}$ oxidation, with the process finishing at around 400 °C evidenced by a plateau on the mass curve. The carbon burning process starts at ~300 °C and finishes at ~600 °C according to the mass-spectrometry measurements (Figure S3a–d).

Both thickness and uniformity of the carbon coating strongly depend on the carbon content for the PAN-samples. As it could be analyzed from STEM images presented in Figure 3, increasing the carbon content leads to a more uniform conductive layer coverage up to 4% of carbon content. Further rise of carbon content to 8% results in the appearance of a freestanding carbon, which could create an additional charge transfer resistance of the composite. For each sample, the thickness of the carbon layer is not fully uniform, with its variation not exceeding 1 nm. Comparing the samples with the close carbon content (PAN-4 and Glu), a negligible difference in the conductive layer thickness, as well as the uniformity of carbon coating thickness, should be noted. The presence of the freestanding carbon could be the reason for poorer cycling stability of the composite since the electronic conductivity is not uniform along the surface. Additionally, a carbon layer of a larger thickness could serve as a barrier for $Li^+$ diffusion increasing the diffusion length and decreasing the electrolyte penetration [16,32]. Introducing up to 8% of carbon results in less than 1 nm thicker carbon coating, but the presence of a freestanding carbon on the particles surface seems to be more significant as it could be a local point of higher diffusion length, thus reducing $Li^+$ diffusion coefficients and increasing charge transfer resistances.

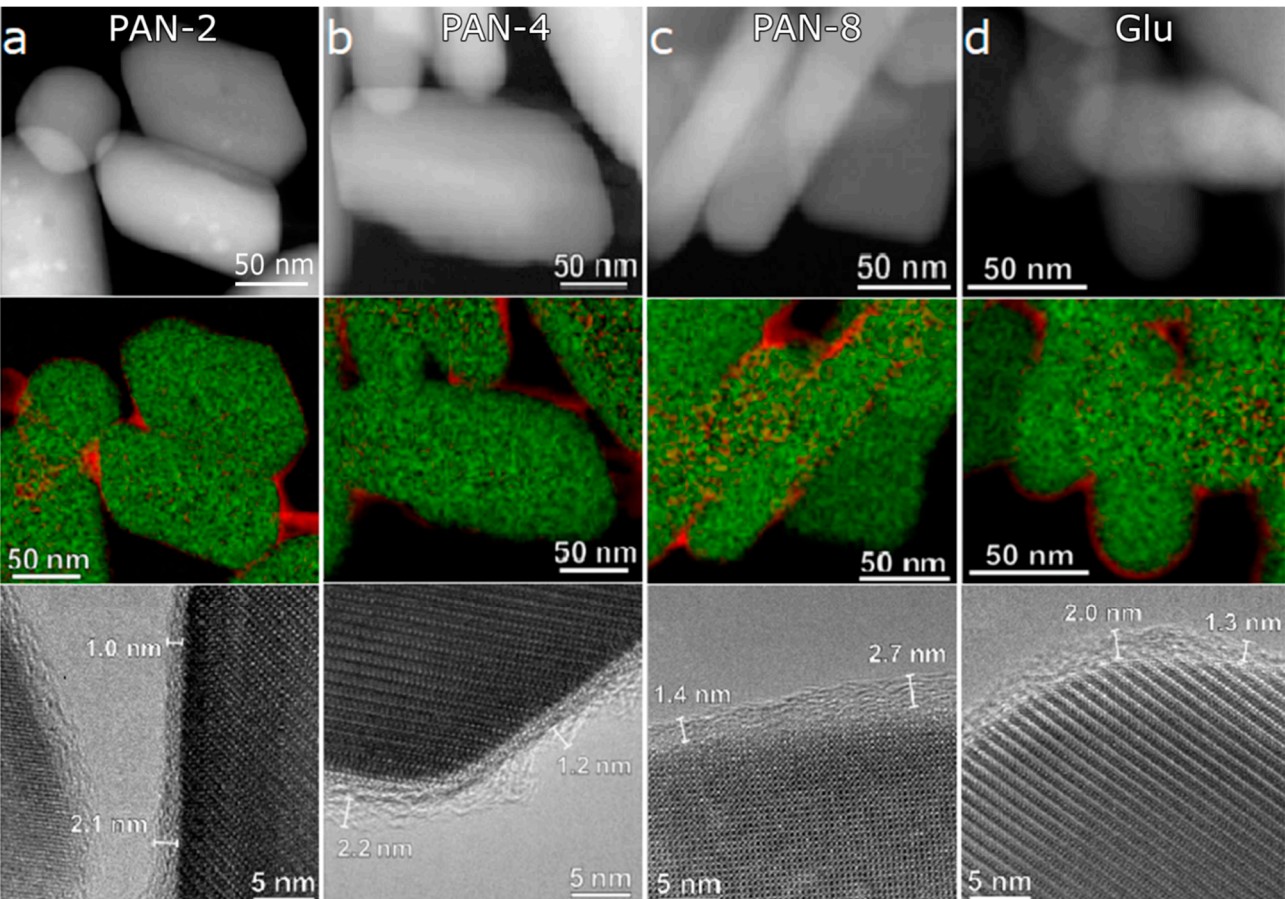

**Figure 3.** DF-STEM images and colored STEM-EELS maps of PAN-2 (**a**), PAN-4 (**b**), PAN-8 (**c**), and Glu (**d**); red color represents the EELS signal from carbon layer, while green color is overall inorganic signal from the sample.

The types of organic functional groups contained in the Li-rich LFMP covered with carbon coating precursors are depicted in Figure 4b. For both PAN- and Glu-coated samples, a signal from the (C=C) group could be detected. For the PAN sample, there are visible signals from ~1660 and ~1450, corresponding to (C=N) and (C-H) bond bending, correspondingly [33], and, after the annealing process in PAN-derived graphitic carbon, there is a broad signal from C-N bonds, while for the Glu sample the intensity of C=C bonds is more pronounced due to a more graphitic nature of the annealed carbon (Figure S4). The graphitization degree of the resulting composite is represented in Figure 4a. Analyzing the D and G band distribution, PAN-derived composites have a more pronounced D band, and the conductive layer could be characterized as mostly consisting of the amorphous carbon. The opposite result is observed for the Glu-derived sample, where the graphitic carbon forms the conductive layer. The $I_D/I_G$ values for the PAN samples are equal to 1.4 for PAN-2, 1.3 for PAN-4, and 1.2 for PAN-8, and, for the Glu sample, the ratio is 0.9. The additional signal at ~950 cm$^{-1}$ from the PO$_4$ group is evidenced for all the samples.

The charge–discharge profiles of the Li-rich LFMP/C composites at low current densities depicted in Figure 5 reveal two plateaus typically assigned to the Fe$^{2+/3+}$ and Mn$^{2+/3+}$ transitions at lower and higher potentials, respectively. At high C-rates, a third intermediate plateau appears for all the samples, potentially leading to a non-uniform reduction of Mn$^{3+}$ and Fe$^{3+}$. The possible explanation was suggested in [34], where the reduction of Mn$^{3+}$ at the potential close to 3.6 V continues in the presence of partially reduced Fe$^{3+}$, which serves as a reducing agent for Mn$^{3+}$ transformation to Mn$^{2+}$. Considering the effect of the carbon content on the shape of the charge–discharge profiles, one could notice a higher polarization for PAN-2 than for PAN-4 and PAN-8, but the values of discharge capacity delivered at low

C-rates for PAN-2 are approximately 5% higher, as well as specific energy values, compared to PAN-4, which makes this particular Li-rich LFMP/C composite suitable for low current densities applications, implying that the extracted capacity of the material is preferential over power.

a
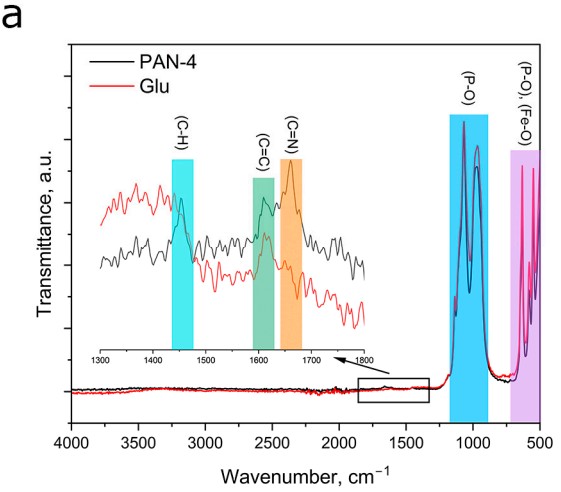

b
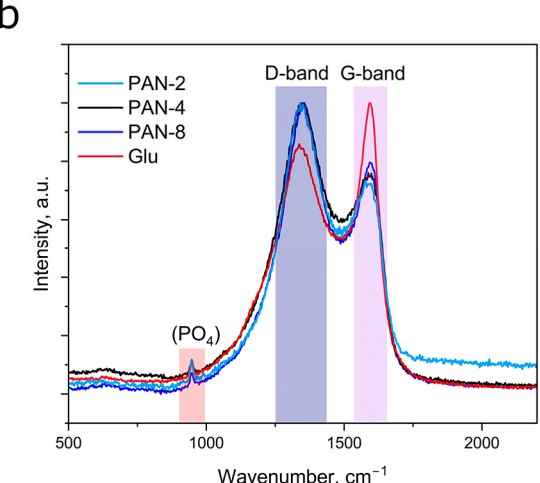

**Figure 4.** Raman spectra of PAN and Glu samples representing D and G band distribution of the composites (**a**) and IR spectra of PAN- and Glu-coated Li-rich LFMP samples with magnified region of organic functional groups signals (**b**).

Comparing PAN-4 and Glu as the samples with the close carbon content and conductive layer thickness, the only difference between the samples is the charge transfer resistance ($R_{ct}$), retrieved from the impedance spectroscopy experiment (Figures 6 and S5). The semicircle in the high frequency region could be assigned to the resistance of cathode/electrolyte interface (CEI), while the medium-frequency semicircle was attributed to $R_{ct}$ [35]. In the $Fe^{2+/3+}$ transition region, the medium frequency semicircle is poorly resolved due to the lower $R_{ct}$ and overlap with the Warburg region of the spectra, while during the $Mn^{2+/3+}$ transition, the lower frequency arc is clearly distinguishable, and the $R_{ct}$ values are three orders of magnitude higher compared to those for the Fe redox region. For both samples, the $R_{ct}$ values increase as the delithiation process approaches the $Li_{0.5}Fe_{0.5}Mn_{0.5}PO_4$ composition and decrease as the delithiation proceeds further. The overall tendency within the samples demonstrates a higher resistivity of the Glu coating for both Fe- and Mn-related transitions: in the Fe redox region, $R_{ct}$ for the PAN is twice lower than that for the Glu sample, while in the case of Mn, where typically the kinetics are much more sluggish [36], the difference is more dramatic, and the Glu-derived carbon coating appears five times more resistive.

Analyzing the high C-rate performance and cycling stability for PAN-4, 8, and Glu, depicted in Figure 7, one could notice a lower output capacity of PAN-4, but higher cycling stability. PAN-8 and Glu have almost the same electrochemical response at 5C, albeit different capacity fade profiles at 10C. In contrast to PAN-4, Glu demonstrates a higher discharge capacity until the 400th cycle, but after that, a dramatic capacity fade reproducibly occurs. Thus, PAN-8 is more stable than Glu and delivers decent electrochemical response even after 500 cycles, but then capacity retention of the sample starts decaying fast. PAN-4 capacity retention could be characterized as a linear decrease in the discharge capacity. The material exhibits 98 ± 4 mAh/g after 1000 cycles of the combined 5C and 10C cycling protocol, which is 27% higher than PAN-8 and 40% higher than the Glu sample.

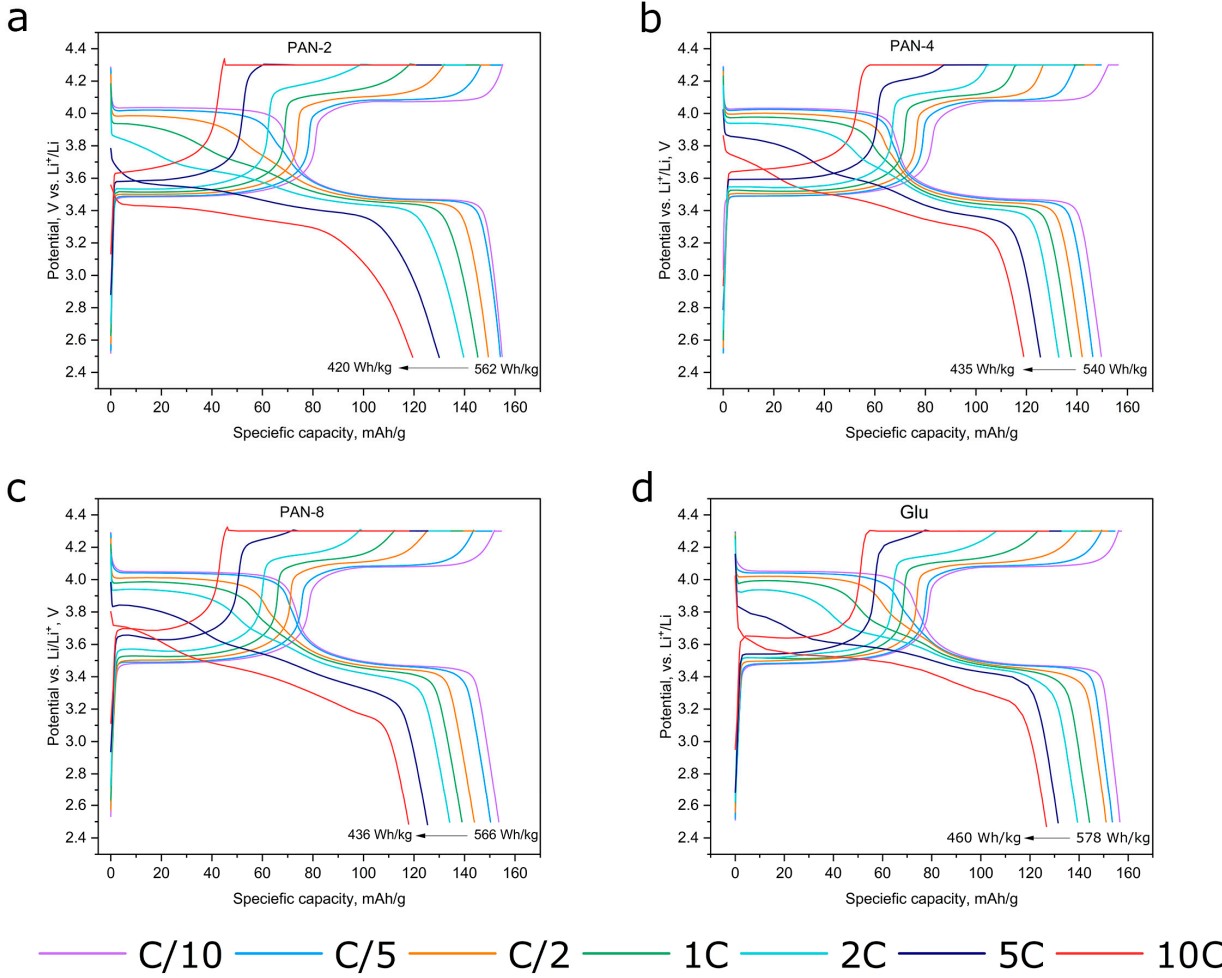

**Figure 5.** Charge–discharge profiles at C/10–10C rates of PAN-2 (**a**), PAN-4 (**b**), PAN-8 (**c**) and Glu (**d**).

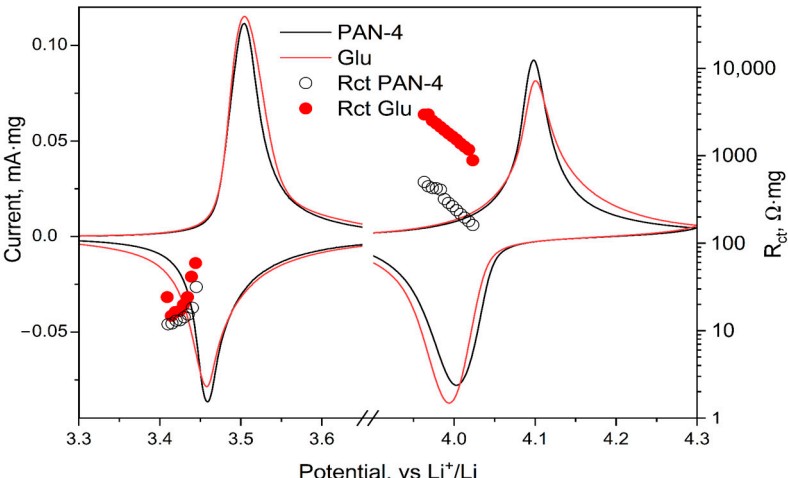

**Figure 6.** The charge transfer resistance in the single-phase regions and CVs of PAN-4 and Glu samples.

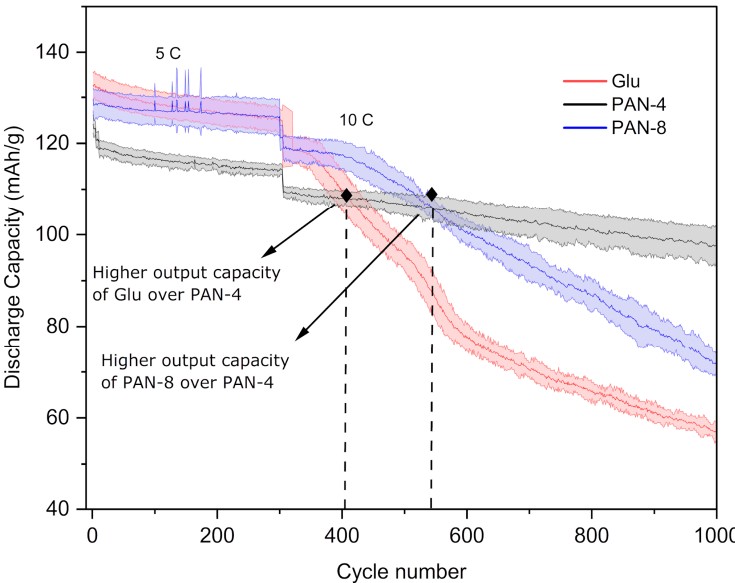

**Figure 7.** The comparison among PAN-4, PAN-8 and Glu samples stability at high current densities; the filled area corresponds to the standard deviation obtained from three coin-cells measurements.

## 4. Discussion

According to the above-mentioned experimental results, Li-rich LFMP/C composites could be designed for a particular application suitable for certain cycling protocols by simply varying the carbon coating precursor or carbon content in the case of PAN samples. The samples with lower carbon content (PAN-2, Glu) are suitable for high output specific capacity at low current densities. The composites with higher carbon content (PAN-4) could be used for high-power application exhibiting beneficial cycling stability, but slightly less extracted specific capacity. Introducing up to 8% of carbon significantly reduces the cycling stability due to a higher carbon layer thickness and the presence of freestanding carbon, which possibly induces greater charge transfer resistances. Comparing the Glu and PAN samples with close carbon content and conductive layer uniformity, one could notice a dramatic difference in the charge transfer resistance based on the impedance spectroscopy experiment. The possible explanation is that an amorphous conductive layer derived from PAN at thermal decomposition, due to the absence of a regular structure, has a lower hardness and better adhesion to the material's surface, which helps sustain a larger mechanical stress during volume expansion/reduction at high current densities [37]. Additionally, the presence of CN groups of the graphitized carbon could be responsible for high LFMP surface affinity and charge transfer promotion [20].

Controlling the carbon content among PAN samples makes it possible to manage the output specific capacity, as well as cycling stability of the material, which leads to the production of the material suitable for a particular application: a high-capacity material for low current loads applications (PAN-2) or a long-life material withstanding high C-rates over hundreds of cycles (PAN-4-8). This finding, along with the other modifications of triphylite-type cathodes, opens up new possibilities of controlling production of the materials for high-capacity or high-power applications. Further research in the area of electrochemical performance improvement of nanosized cathode materials may include N- and O- doping of PAN-derived carbon coating in order to increase the electronic conductivity of the resulting composite. One more direction that is worth mentioning is increase in the extracted specific capacity since the PAN-derived Li-rich LFMP/C composite with high carbon content (≥4% mass) demonstrates lower electrochemical performance, and the reason could be in hindered $Li^+$ diffusion. In that sense, the coating technology should be modified for producing thinner carbon-coated nanosized materials. Another modification of the suggested technology could be an application of mixed carbon precursors in

order to investigate the influence of mechanical and electronic properties on the overall electrochemical performance of the material.

Additionally, a combined carbon coating made by PAN and Glu could be a promising strategy, since it could synergize with the positive features of both carbon coating precursors: lower charge-transfer resistance of PAN and higher output capacity of Glu. Low carbon content and the absence of other conductive additives during the composite preparation makes the utilization of PAN attractive in terms of preserving the gravimetric capacity of the material, since there is a negligible amount of electrochemical inactive species in the composite.

Additional research on the influence of the pre-oxidation process of PAN on electronic conductivity seems to be promising, since it potentially leads to the reduction of carbon black added to the electrode slurry during the electrode preparation and to the increase in tap density of the resulting composite. Additionally, the use of PAN as a carbon source and as a binder material could increase the adhesion of composite on the electrode foil due to the presence of C–N bonds on the surface of electrode material and the binder. With proper electronic conductivity and mechanical properties of the obtained thin film, a manufacturing of free-standing electrodes without metal foil could be possible.

### 5. Conclusions

The application of PAN as a carbon coating precursor allows one to improve the cycling stability of the Li-rich LFMP/C at high current densities, and the material with 4% of carbon content demonstrates especially attractive electrochemical performance and exhibits more 112 mAh/g with capacity retention of $78 \pm 2\%$ after sever cycling protocol at 5C and 10C. As a comparison, a conventional glucose-derived carbon coating demonstrates only $43 \pm 3\%$ in the same conditions. That makes PAN a promising carbon coating precursor for Me-ion cathode materials designed for high power application. The PAN-derived carbon coating technology could be suggested to other cathode materials with low electronic conductivity, which makes the approach versatile and efficient in terms of preserving the decent output capacity during long cycling.

**Supplementary Materials:** The following supporting information can be downloaded at: https://www.mdpi.com/article/10.3390/applnano4010002/s1, Figure S1: SEM image of PAN particles; Table S1: Mass ratio for producing Li-rich LFMP/C composites; Figure S2: Experimental TG curve for pristine Li-rich LFMP; Figure S3: Experimental TG curve for PAN-2 (a), PAN-4 (b), PAN-8 (c) and Glu (d); Figure S4: Organic functional groups presenting in graphitized PAN and Glu after 650 °C; Figure S5: Impedance spectra of LFMP/C in the single-phase regions of Fe redox for Glu (a) and PAN-4 (b) and Mn redox for Glu (c) and PAN-4 (d). Symbols are experimental data, lines represent the fitted impedance spectra; Figure S6: Current transients of PAN-4 sample assigned to the diffusion and two-phase transients before and after the phase transition of Fe (a) and Mn (b); Figure S7: Equivalent circuit model for impedance data treatment, where $R_{sol}$ represents the resistivity of electrolyte, $R_{SEI}$ and $CPE_{SEI}$ are resistivity and capacitance of SEI layer, $R_{ct}$ and $CPE_{dl}$—charge transfer resistance and double layer capacitance, W—Warburg element.

**Author Contributions:** Conceptualization, E.E.N., E.V.A. and S.S.F.; methodology E.E.N. and O.A.T.; validation, E.E.N., O.A.T., V.A.N., E.V.A. and S.S.F.; formal analysis, E.E.N., V.A.N., E.V.A. and S.S.F.; investigation, E.E.N. and O.A.T.; data curation, E.E.N. and O.A.T.; writing—original draft preparation, E.E.N.; writing—review and editing, O.A.T., V.A.N., E.V.A. and S.S.F.; visualization, E.E.N., O.A.T., V.A.N., E.V.A. and S.S.F.; supervision, S.S.F.; project administration, S.S.F.; funding acquisition, S.S.F. All authors have read and agreed to the published version of the manuscript.

**Funding:** This research was funded by the Russian Foundation for basic research, grant number 20-33-90291.

**Data Availability Statement:** All data supporting these findings are presented in the manuscript and supporting information. Any other data might be requested from the authors upon reasonable request.

**Acknowledgments:** AICF of Skoltech is acknowledged for allocating time to perform TEM measurements. The authors are grateful to Maria Kirsanova for conducting STEM and EELS analyses.

**Conflicts of Interest:** The authors declare no conflict of interest. The funders had no role in the design of the study, in the collection, analyses, or interpretation of data, in the writing of the manuscript, or in the decision to publish the results.

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
