# Peer review of "Polyacrylonitrile-Derived Carbon Nanocoating for Long-Life High-Power Phosphate Electrodes"

_2673-3501, doi:10.3390/applnano4010002_

Round 1
Reviewer 1 Report
This work explored the effect of polyacrylonitrile-(PAN)-derived carbon coating on the electrochemical properties of the cathode materials. The modified electrode materials show a good performance and obtain a high-capacity or high-power performance by varying the carbon content. This is an interesting work and a good contribution to Applied Nano. Following revisions are suggested to improve the quality of this article:
1. What are the advantages of the proposed PAN-derived carbon coating in this manuscript compared to other carbon sources (such as MOF materials) reported in the literature?
2. What is the effect of carbon coating with different carbon contents on Li+ diffusion?
3. Some recent articles on carbon coatings for Li-ion batteries (such as doi: 10.1021/acsami.2c03813; 10.1039/D2QI01973F) are suggested to be referred.
4. The paper needs to be carefully proofread and English polished to ensure a proper readability.
Author Response
Response to Reviewer 1
Point 1: What are the advantages of the proposed PAN-derived carbon coating in this manuscript compared to other carbon sources (such as MOF materials) reported in the literature?
Response 1: We added a passage devoted to the MOF utilization as a carbon coating precursor for electrode materials. As in the case of PAN application, the described technology allows controlling the surface properties of the obtained carbon coating, but the whole process of using MOFs seems to be more complicated and less energy efficient that the proposed PAN technology.
“Another approach allowing tailoring the surface properties of the obtained carbon coating is using MOFs (Metal organic frameworks) as a carbon coating precursor [26]. Like for PAN technology it is possible to control the electronic conductivity of the carbon layer by the introduction of additives or engineering the chemical structure of organic framework [27-28]. Currently, the described approach is applied mostly for a laboratory application due to the complication of a synthesis procedure of MOF materials. Usually, the process requires a number of steps including solvothermal preparation, solvent extraction from the porous structure and careful drying process at high temperature [29].”
Point 2: What is the effect of carbon coating with different carbon contents on Li+ diffusion?
Response 2: We provided a few references to the existed research on the topic. Indeed, the thicker carbon layer could be a barrier for Li+ diffusion, in our case the conductive layer thickness varies negligibly with the increase of carbon content, so the presence of a freestanding carbon could locally reduce the diffusion coefficient and increase the charge transfer resistance of the composite.
“Also, a carbon layer of a larger thickness could serve as a barrier for Li+ diffusion increasing the diffusion length and decreasing the electrolyte penetration [16,32]. Introducing up to 8% of carbon results in less than 1nm thicker carbon coating, but the presence of a freestanding carbon on the particles surface seems to be more significant as it could be a local point of higher diffusion length thus reducing the Li+ diffusion coefficient and increasing charge transfer resistance.”
Point 3: Some recent articles on carbon coatings for Li-ion batteries (such as 10.1021/acsami.2c03813; 10.1039/D2QI01973F) are suggested to be referred.
Response 3: Thank you for the suggestion, but unfortunately the provided papers are out of the scope of our paper. The papers describe the application of carbon black as a conductive agent to metal phosphides as anode materials for Na-ion batteries, while our paper is focused on the carbon coating application during the decomposition process of the polymer on the surface of a cathode material for Li-ion batteries. The references to SnO2 and GeP materials were made to illustrate the versatility of the PAN coating technology.
Point 4: The paper needs to be carefully proofread and English polished to ensure a proper readability.
Response 4: The manuscript was carefully polished, all the misprints and mistakes were corrected. Please, find all the corrections in the reviewed file.
Reviewer 2 Report
The manuscript corresponds to the Applied Nano. The Introduction (Lines 25-111) and the list of references (Lines 386-482) are quite complete. The methodology of the study is described in sufficient detail, however, it can still be improved.
The text is reasonably clear and easy to read.
All structural units of the manuscript are logically interconnected, except the conclution.
The manuscript contains important scientific results for practice. Therefore, the manuscript is of interest to many specialists in this and related fields.
Comments and suggestions:
1. Line 38: “the de/intercalation mechanism [2], [6], [7];” , the most appropriate way is “the de/intercalation mechanism [2, 6-7];”, I suggest they be changed where they exist.
2. Ηave you checked how oxidation affects your results?
3. The conclusion is missing from the article, it should be included.
Author Response
Response to Reviewer 2
Point 1: Line 38: “the de/intercalation mechanism [2], [6], [7];”, the most appropriate way is “the de/intercalation mechanism [2, 6-7];”, I suggest they be changed where they exist
Response 1: We took into account the suggestion and corrected the references accordingly.
“the diffusion network dimensionality [4-5]”
“the de/intercalation mechanism [2, 6-7]”
“dissolution of Mn2+ in the electrolyte [8-9]”
“utilized: polyacrylonitrile (PAN) [14-15]”
“structure of the organic framework [27-28]”
Point 2: Ηave you checked how oxidation affects your results?
Response 2: According to the FTIR and Raman spectroscopy results, the PAN-derived graphitized carbon does not contain any C-O functional groups, but the effect of oxidation on electronic conductivity of the resulted carbon coating seems to be a valuable tool for tuning the surface properties of the composites. In the discussion we stated it as our further direction of research.
“Additional research on the influence of the pre-oxidation process of PAN on electronic conductivity seems to be promising since it potentially leads to the reduction of carbon black added to the electrode slurry during the electrode preparation and to the increase of tap density of the resulting composite. Also, the use of PAN as a carbon source and as a binder material could increase the adhesion of composite on the electrode foil due to the presence of C-N bonds on the surface of electrode material and the binder. With proper electronic conductivity and mechanical properties of the obtained thin film a manufacturing of free-standing electrodes without metal foil could be possible.”
Point 3: The conclusion is missing from the article, it should be included.
Response 3: We added the conclusion to the main text.
- Conclusion
The application of PAN as a carbon coating precursor allows to improve the cycling stability of the Li-rich LFMP/C at high current densities, the material with 4% of carbon content demonstrates especially attractive electrochemical performance and exhibits more 112 mAh/g with capacity retention of 78±2% after sever cycling protocol at 5C and 10C. As a comparison, a conventional glucose-derived carbon coating demonstrates only 43±3% in the same conditions. That makes PAN a promising carbon coating precursor for Me-ion cathode materials designed for high power application. The PAN-derived carbon coating technology could be suggested to other cathode materials with low electronic conductivity, which makes the approach versatile and efficient in terms of preserving the decent output capacity during long cycling.
Round 2
Reviewer 1 Report
The authors have put considerable effort into addressing the reports of the referees. As a result, the submission has been greatly improved and is worthy of publication.